# Feasibility of non-contact cardiorespiratory monitoring using impulse-radio ultra-wideband radar in the neonatal intensive care unit

**Won Hyuk Lee[1], Yonggu Lee[2], Jae Yoon Na[3], Seung Hyun Kim[3], Hyun Ju Lee[3], Young-Hyo Lim [2], Seok Hyun Cho[4], Sung Ho Cho [1]\*, Hyun-Kyung Park [3]\***

**1** Department of Electronics and Computer Engineering, Hanyang University, Seoul, Republic of Korea, **2** Division of Cardiology, Department of Internal Medicine, Hanyang University College of Medicine, Seoul, Republic of Korea, **3** Department of Pediatrics, Hanyang University College of Medicine, Seoul, Republic of Korea, **4** Department of Otorhinolaryngology, Hanyang University College of Medicine, Seoul, Republic of Korea

☯ These authors contributed equally to this work.
\* neopark@hanyang.ac.kr (HKP); dragon@hanyang.ac.kr (SHC)

## Abstract

### Background

Current cardiorespiratory monitoring equipment can cause injuries and infections in neonates with fragile skin. Impulse-radio ultra-wideband (IR-UWB) radar was recently demonstrated to be an effective contactless vital sign monitor in adults. The purpose of this study was to assess heart rates (HRs) and respiratory rates (RRs) in the neonatal intensive care unit (NICU) using IR-UWB radar and to evaluate its accuracy and reliability compared to conventional electrocardiography (ECG)/impedance pneumography (IPG).

### Methods

The HR and RR were recorded in 34 neonates between 3 and 72 days of age during minimal movement (51 measurements in total) using IR-UWB radar ($HR_{Rd}$, $RR_{Rd}$) and ECG/IPG ($HR_{ECG}$, $RR_{IPG}$) simultaneously. The radar signals were processed in real time using algorithms for neonates. Radar and ECG/IPG measurements were compared using concordance correlation coefficients (CCCs) and Bland-Altman plots.

### Results

From the 34 neonates, 12,530 HR samples and 3,504 RR samples were measured. Both the HR and RR measured using the two methods were highly concordant when the neonates had minimal movements (CCC = 0.95 between the $RR_{Rd}$ and $RR_{IPG}$, CCC = 0.97 between the $HR_{Rd}$ and $HR_{ECG}$). In the Bland-Altman plot, the mean biases were 0.17 breaths/min (95% limit of agreement [LOA] -7.0–7.3) between the $RR_{Rd}$ and $RR_{IPG}$ and -0.23 bpm (95% LOA -5.3–4.8) between the $HR_{Rd}$ and $HR_{ECG}$. Moreover, the agreement for the HR and RR measurements between the two modalities was consistently high regardless of neonate weight.

**Data Availability Statement:** All relevant data are within the manuscript and its Supporting Information files.

**Funding:** This research was supported by the Korea Special Therapeutic Education Center (Chairman Il-Kewon Kim) of Anyang, Korea, and the Bio and Medical Technology Development Program (Next Generation Biotechnology) through the National Research Foundation of Korea (NRF) funded by the Ministry of Science, ICT and Future Planning (NRF-2017M3A9E2064735).

**Competing interests:** The authors have declared that no competing interests exist.

## Conclusions

A cardiorespiratory monitor using IR-UWB radar may provide accurate non-contact HR and RR estimates without wires and electrodes for neonates in the NICU.

## Introduction

The most widely used cardiorespiratory monitoring technologies in the neonatal intensive care unit (NICU) are pulse oximetry based on photoplethysmography, electrocardiography (ECG) and impedance pneumography (IPG) based on electrical potential differences obtained through adhesive electrodes on the skin. However, IPG suffers from inaccuracy and cardiac interference in neonates with rapid respiratory rates (RRs) and limited lung aeration because it is based on breath-dependent thoracic variations in transthoracic impedance [1–3]. Moreover, these instruments have several additional disadvantages resulting from the use of adhesive sensors. Repetitive replacement of electrodes and the twining wires around the arm or leg cause skin damage, infections due to skin layer breakdown, permanent scars, and circulatory disturbances, particularly in premature infants with fragile skin. There may even be a risk of hypothermia during procedures, which could cause circulatory disturbances, particularly in premature infants.

In recent years, significant attention has been paid to non-contact novel methods for vital sign assessment in neonates [1, 2, 4–9]. However, these studies either were explorative with small sample sizes or reported on techniques used to monitor only RR or HR (heart rate) [3, 10–13]. Trials for contactless cardiorespiratory simultaneous measurement have not been reported in the NICU population. Results obtained from infants and adults are limited and can only influence clinical practices for neonates and preterm infants against a reference gold standard in terms of feasibility, accuracy, and standardization.

Impulse radio ultra-wideband radar (IR-UWB radar) is a high-precision electromagnetic sensor that recognizes the motion of an object at a distance. IR-UWB radar has various advantages in medical applications, such as its contactless/wireless use, license-free use, easy application, low cost, high data-processing rate, low exposure risk for the human body, and daily convenient use in and out of the hospital [14–17]. Accurately monitoring neonatal HR is important for clinicians to assess the well-being of neonates in the NICU. Recently, we presented an RR monitoring algorithm for IR-UWB radar to extract the breathing signal and demonstrated the feasibility and accuracy of radar as an RR monitor for neonates [18]. However, accurate HR estimation using radar is still challenging because neonate HR often reaches more than twice the adult HR. Moreover, the signal intensity from the neonate's heart is weaker, and the harmonics from the neonate's rapid breathing are hindered. To improve the quality of HR assessment in neonates using our radar technology, we redesigned the data-processing algorithms for the radar signals used in adults [19–21] and finally achieved sufficient accuracy for HR measurements using the radar in neonates.

The aim of this study is to investigate the performance of simultaneous non-contact measurements for both HR and RR using our IR-UWB radar technology compared to that of conventional ECG/IPG monitors in the NICU.

## Materials and methods

### Subjects

From July 2018 to February 2019, we prospectively enrolled 34 neonates (16 preterm and 18 term babies) from the NICU in Hanyang University Hospital, Seoul, Korea. A total of 51

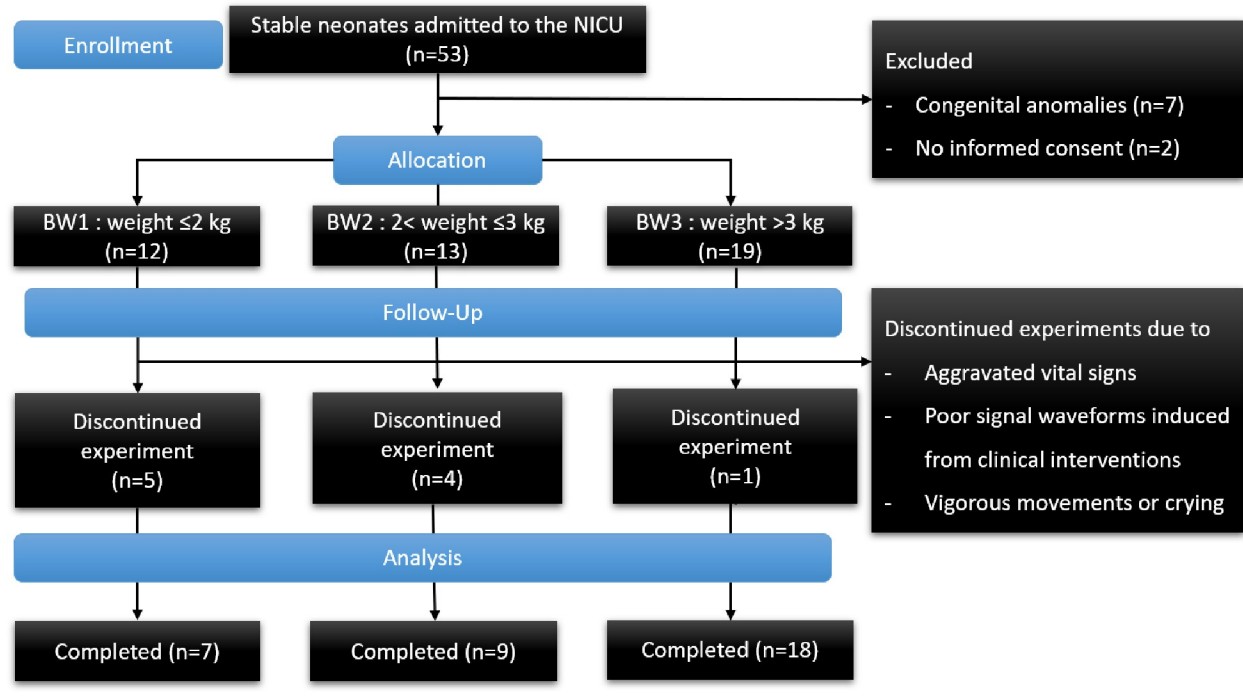

**Fig 1. CONSORT flow diagram.**

measurements were obtained from those 34 neonates and were used to compare the accuracy of the ECG/IPG and the IR-UWB radar. Among the 34 neonates, 33 neonates breathed spontaneously without supplemental oxygen and invasive/non-invasive respiratory assistance, while one preterm neonate received synchronized intermittent mechanical ventilation therapy during measurement. Neonates with congenital anomalies or unstable conditions, including hypotension, sustained tachypnea (RR > 60 breaths/min) and fever (>38˚C), were excluded from the study because of the need for frequent medical care and intervention (Fig 1). The study protocol adhered to the Declaration of Helsinki, approved by the Institutional Review Board of Hanyang University Medical Centre (No. 2017-09-046-002) and registered in ClinicalTrials.gov (NCT03622996). Written informed consent was obtained from the parents.

## Experimental setup

All experiments were conducted at the bedside in the NICU. The IR-UWB radar and a conventional vital sign monitor using ECG/IPG were measured simultaneously. The radar chip was covered with a plastic cap and was placed at the end of a flexible arm on a tripod, which was approximately 1 metre in height from the floor, pointing at the chest of each neonate. The neonates were laid inside an open-air crib or incubator in a supine position, and their torsos were covered with a blanket. The radar was placed at a distance of 35 cm and was orthogonal to the chest (Fig 2). The cradles or incubators were fixed from motion during the experiment. The measurement of radar was obtained when the neonates were left alone. Clinical workflow always took priority over the measurement, and whenever a medical procedure was required, the measurement was temporarily suspended. The data obtained from the radar were processed and stored in a laptop computer placed in the vicinity.

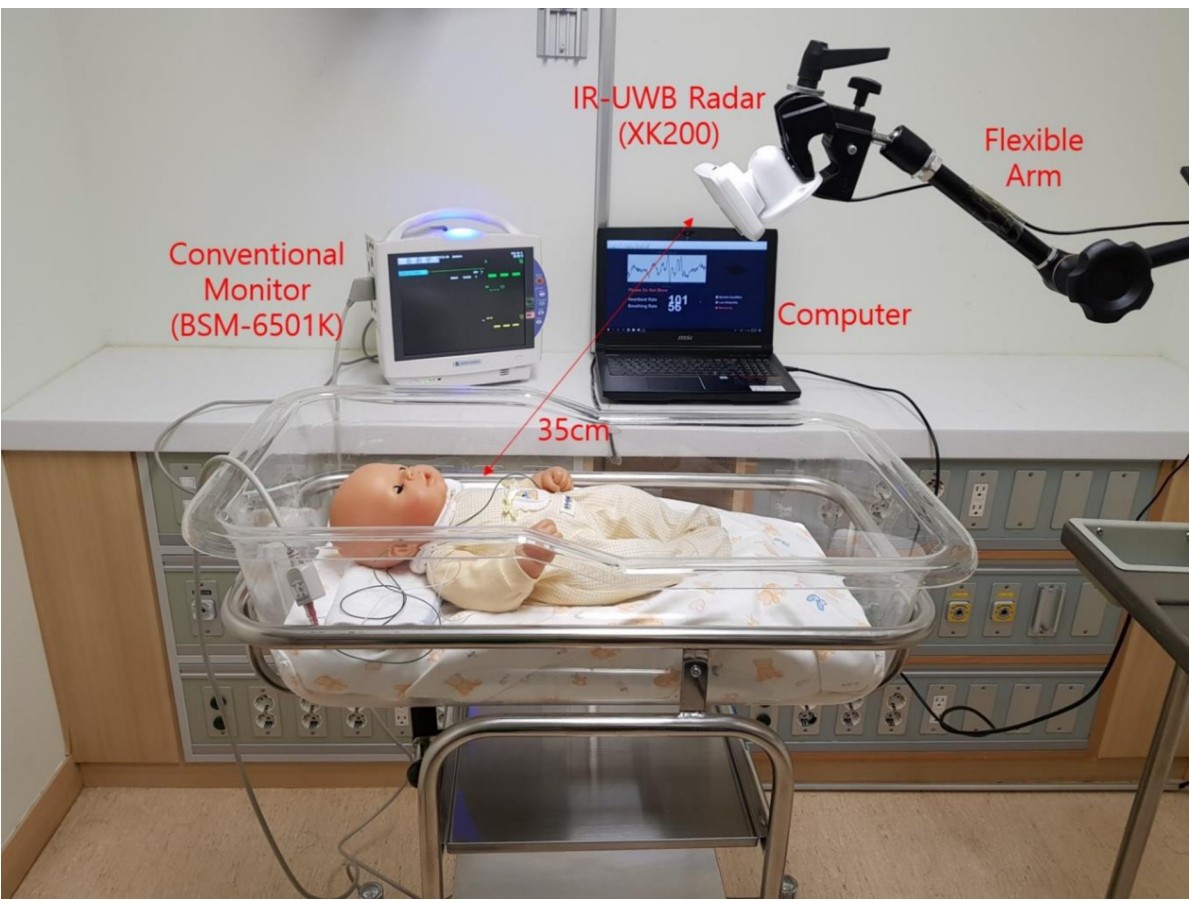

**Fig 2. Experimental set-up for simultaneous IR-UWB radar and ECG/IPG recording.** The radar sensor was covered with a cap (width × depth × height, 5.8 × 3.4 × 1.8 cm, weight, 150 g; inside the actual sensor chip: 2.2 × 1.2 × 0.6 cm, 18 g), placed on an arm attached to the cradle and pointed at the chest of the neonate at a perpendicular angle. The sampling rate of the radar measurement was 60 Hz. The BSM-6501K monitor (Nihon Kohden, Tokyo, Japan) is used as the reference monitor for both HR and RR. The neonate's clothes remain on during measurement.

## Conventional monitor (ECG/IPG) measurement

A BSM-6501K patient monitor (Nihon Kohden, Tokyo, Japan) was used as a reference monitor. The three transcutaneous electrodes were attached at the standard positions, and a pulse oximetry sensor was placed on the sole of the neonates. HR measured using ECG ($HR_{ECG}$) was calculated with the last 12 consecutive heartbeat intervals and the RR measured using IPG ($RR_{IPG}$) was calculated with the last 8 respiration intervals. The measurements were recorded on an external storage device every second and extracted using viewer software (BSM Viewer, Nihon Kohden, Tokyo, Japan). The values of $HR_{ECG}$ were averaged over every 10 seconds to compare with the radar measurements.

## Radar data collection and processing

A commercially available IR-UWB radar device, XK200 (Xandar Kardian, Delaware, USA), was used to send and collect radar signals to and from the chest. MATLAB (MathWorks, Natick, MA, USA), a commercially available software package, was used to acquire, process and store the data from the radar sensor. The IR-UWB radar operates within an FCC mask (US Federal Communications Commission Mask Regulation) [20, 22–24], and its safety as

## Algorithm Block diagram

**Fig 3. Algorithm block diagram.** HR and RR detection algorithm for neonates redesigned from the algorithm used for adults. Because the raw signals received from the radar contained noise components, signal processing algorithms were applied to the raw signal. After the breathing waveform and its harmonics were removed using a band-pass filter, the HR frequency components were estimated using fast Fourier transform.

low-power wireless equipment was certified by the National Radio Research Agency, Ministry of Science, ICT and Future Planning, Korea (certification no. MSIP-CRM-Top-TSR-M200W). The certification can be found on the following website (https://www.xkcorp.com/certifications). Signals obtained from the radar were transferred to a computer for processing and frequency analysis (Fig 3).

Because the measurements using the radar as well as those using ECG/IPG could significantly be interfered by large movements on both limbs and the torso, we quantified spontaneous or medical care-related body movements of neonates with the power differences in radar measurements to identify certain notable body movements using radar. The notable body movements included being in nursing care, repetitive myoclonus, hiccupping, flopping and crying; the other movements were considered minimal movements. Neonates were recorded using a video camera during the entire radar measurements, and the movements in the video footages were then compared with the quantified movements in the radar to produce a cut-off criterion for those notable movements [25] (S1 Data).

Fast Fourier Transform (FFT) was used to convert frequency domains in the extracted radar signals into the $RR_{Rd}$ (RR measured using the IR-UWB radar) and $HR_{Rd}$ (HR measured using the IR-UWB radar). $RR_{Rd}$ was derived from the average of the frequencies with the largest magnitude within the RR range over 10 seconds. Unlike $RR_{Rd}$, the frequency component of $HR_{Rd}$ can be identified with its lower magnitude and higher range compared with those of the $RR_{Rd}$ component [20, 24]. Because the frequency components of $HR_{Rd}$ were similar to the

harmonic frequency components of $RR_{Rd}$ in both magnitude and range, $HR_{Rd}$ was obtained through a harmonic cancellation algorithm to suppress breath harmonics within the $HR_{Rd}$ frequency domains (S1 Fig).

The radar tries to observe the vital sign signal through the movement near the subject's abdomen, but the movement hinders acquisition of these vital signals. Therefore, the algorithm tries to remove as much movement/artifact components as possible from the radar signals received. However, the RR and HR extracted during movement are not reliable, similar to data obtained from existing patient monitors.

## Modification of radar measurements for neonates

Because of physiologic and anthropometric differences between neonates and adults, we modified some measurement settings that we had introduced in the previous studies on adults [19, 20] as follows.

1. Lower signal-to-noise ratio (SNR) circumstance: Fixing the observation points of IR-UWB radar to one body part is difficult because neonates and premature babies are much smaller than adults. The observation point is a specific distance point that determines the best vital sign observed on radar to extract. The SNR will be reduced because all vibrations or small movements generated by the treatment will be reflected in the radar signals. Additionally, the HR movements of infants are smaller than those of adults because infants have small bodies, which is why we want to increase the Frames per second (FPS) of the radar to increase the quality of the signal. The 20 FPS was used to extract the vital signs of adults, but 60 FPS was applied for neonates and premature infants.

2. Suitable location for the best detection in the neonates: The capability to detect the motion of the abdomen by breathing does not vary significantly depending on the direction of the radar installation. However, to measure the HR, the signals received by the radar must adequately reflect the movement of the heart, which requires the radar to be positioned vertically from the lying body and not from the head or legs of the infant.

3. Selection of different frequency spectrum ranges: The HR of neonates and premature babies is 90 to 200 beats per minute [26, 27], which is more than twice the average HR of adults. To measure the HRs of neonates and premature babies, a band-pass filter in different bands compared to adults should be designed. A band-pass filter with cut-off frequencies of 10 (breaths/min) and 90 (breaths/min) was used to extract breath, and a band-pass filter with cut-off frequencies of 90 (bpm) and 210 (bpm) was used to extract the HR.

4. Integration of body movement: Neonates and premature babies in the NICU sustain spontaneous movements; thus, the reliability of the RR and HR can be reviewed by measuring the degree of movement based on the distance of the signals received by the radar.

5. Extraction of vital sign signals without disturbance from small movements of the baby: Motion such as lifting the hands or legs can disturb the radar signals and confuse the observation points for retrieving heartbeats. Accordingly, the observation point was fixed to the chest area and not to the extremities to prevent interference from motion.

## Statistical analysis

The data are presented as the median with interquartile ranges or the mean with standard deviation (SD). Because the accuracies of both the IR-UWB radar and the conventional monitors were highly influenced by large movements on the limbs and torso, analyses were conducted

separately when there were notable movements in the neonates. The agreements between the radar and ECG measurements were evaluated using Lin's concordance correlation coefficient (CCC) and Bland-Altman plots with 2.5% and 97.5% limits of agreement (LOA). The significances of biases between the two methods were evaluated using a single-sample t-test.

The neonates were divided into 3 groups according to the body weight on the day of radar recording as follow: ≤2 kg for BW1 group; 2< weight ≤3 kg for BW2 group; >3 kg for BW3 group). The bias levels between the IR-UWB radar and the conventional monitors were compared among the 3 groups using a one-way ANOVA. Measurements of HR and RR were also categorized into 3 levels according to the HR and RR from the conventional monitors as follow (HR1/RR1: HR or RR <5%; HR2/RR2: HR or RR of 5%~95%; HR3/RR3: HR or RR >95% in the distribution) to evaluate the systematic biases residing in the HR and RR measurement data and the agreement levels between the two methods in the extremely low or high measurement levels.

All statistical analyses were performed using the statistical software R version 3.4.0 and its packages epiR and MethComp. A $p < 0.05$ was considered statistically significant.

## Results

The baseline characteristics of the subjects are summarized in Table 1. The median gestational and postnatal ages were 38.6 weeks and 14.5 days, respectively. The median birth weight and

**Table 1. Baseline characteristics of the subjects.**

| Demographics | N = 34 |
|---|---|
| Gestational age, weeks | 38.6 (32.4–39.4) |
| Preterm infants, 10/34 (29.4%) | 31.1 (30.0–32.4) |
| Term infants, 24/34 (70.6%) | 39.1 (38.4–39.8) |
| Birth weight, g | 3,085 (1,690–3,370) |
| Male | 18 (52.9%) |
| Singleton | 25 (73.5%) |
| Small for gestational age (SGA) infant | 4 (11.8%) |
| Birth by caesarean section | 20 (58.8%) |
| Apgar 1 min | 6 (4–8) |
| Apgar 5 min | 8 (7–9) |
| Duration of hospitalization, days | 10 (7–20) |
| Breast milk feeding during hospital stay | 19 (55.9%) |
| Age at measurement, days | 14.5 (7–28) |
| Body weight at measurement, g | 3,020 (2,110–3,550) |
| BW1, 7/34 (20.6%) | 1,880 (1,585–1,890) |
| BW2, 9/34 (26.5%) | 2,430 (2,330–2,760) |
| BW3, 18/34 (52.9%) | 3,525 (3,200–3,800) |
| Respiratory rate, breaths/min | 38.1 (32.0–45.2) |
| BW1 | 48.5 (34–63.4) |
| BW2 | 36.0 (31.2–40.9) |
| BW3 | 38.0 (31.7–44.2) |
| Heart rate, bpm | 134.0 (127.3–140.0) |
| BW1 | 146.0 (135.3–149.0) |
| BW2 | 128.0 (122.0–133.3) |
| BW3 | 134.0 (128.6–139.0) |

Data are presented as N (%) or the median (interquartile range). BW1 group, body weight on recording ≤ 2 kg; BW2, 2 < weight ≤3 kg; BW3, weight > 3 kg.

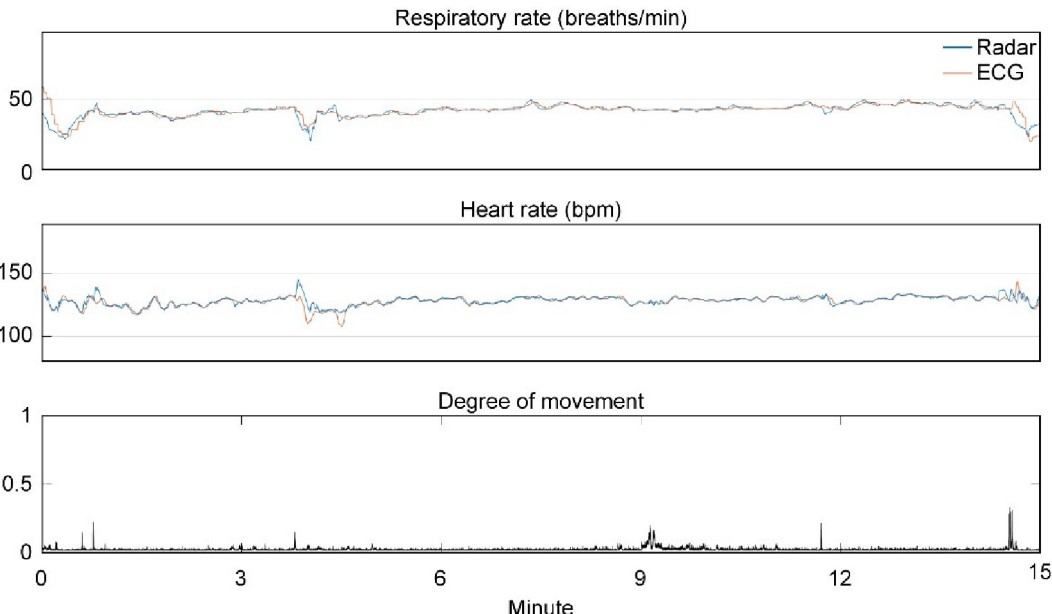

**Fig 4. Representative examples of the HR and RR of a premature infant.** Representative examples of the HR and RR obtained from a 7-day-old premature infant weighing 3,550 g (gestational age 31+2 weeks, birth weight 3,400 g) during sleeping. The blue and red lines represent measurements from the radar ($RR_{Rd}$, $HR_{Rd}$) and ECG/IPG ($RR_{IPG}$, $HR_{ECG}$) over 15 minutes, respectively. The degree of movement is presented with arbitrary units based on the distance from the IR-UWB radar (the lowest panel). The measurements from the two methods agreed well with each other.

body weight were 3,085 g and 3,020 g. The median RR and HR were 38.1 (IQR 32.0–45.2) breaths/min and 134.0 (IQR 127.3–140.0) bpm, respectively. The average total recording time was 44 ± (20.3) min, and the average valid recording time during minimal movement for the final analysis was 22 ± (10.1) min (S1 Table).

A representative case regarding the comparison of HR and RR between the radar and the conventional measurements is depicted in Fig 4. In the tachograms, $HR_{Rd}$ and $RR_{Rd}$ appeared to be highly correlated with $HR_{ECG}$ and $RR_{IPG}$, respectively, and minor discrepancies developed when minimal movements of neonates were present. The measurements using radar showed large discrepancies from those using the conventional methods during the notable movements (S2 Fig).

The comparisons between $RR_{Rd}$ and $RR_{IPG}$ during minimal movements are summarized in Fig 5A–5C. The $RR_{Rd}$ and $RR_{IPG}$ were highly correlated with each other, and the concordance was excellent (CCC 0.95; 95% confidence interval [CI], 0.947–0.954). The Bland-Altman plot shows that the mean bias was significant, whereas it was clinically negligible (0.17 breaths/min; 95% CI, 0.05–0.29; $p<0.001$). The width of the 95% LOA was less than 20% (13.7 breaths/min) of the average RRs at the maximum and gradually decreased with increasing average RRs. The agreement levels were similar among the three BW groups (BW1 group 0.94; 95% confidence interval [CI], 0.934–0.952 vs. BW2 group 0.96; 95% CI, 0.952–0.964 vs. BW3 group 0.94; 95% CI, 0.936–0.946). The mean biases were smallest in the BW3 group, but not significantly different among the groups (Fig 6, S3 Fig, and S1 Table).

The BW1 group indicates <2 kg, BW2 group 2~3 kg and BW3 group >3 kg. The HR1 (or RR1) indicates the measurements with HR (or RR) < lower 5% and the HR3 (or RR3) indicates the measurements with HR (or RR) ≥ upper 5%. Comparisons between $HR_{Rd}$ and $HR_{ECG}$ during the minimal movement are summarized in Fig 5D–5F. Similar to the case of RR, $HR_{Rd}$ and $HR_{ECG}$ were highly correlated with each other, and the concordance level was

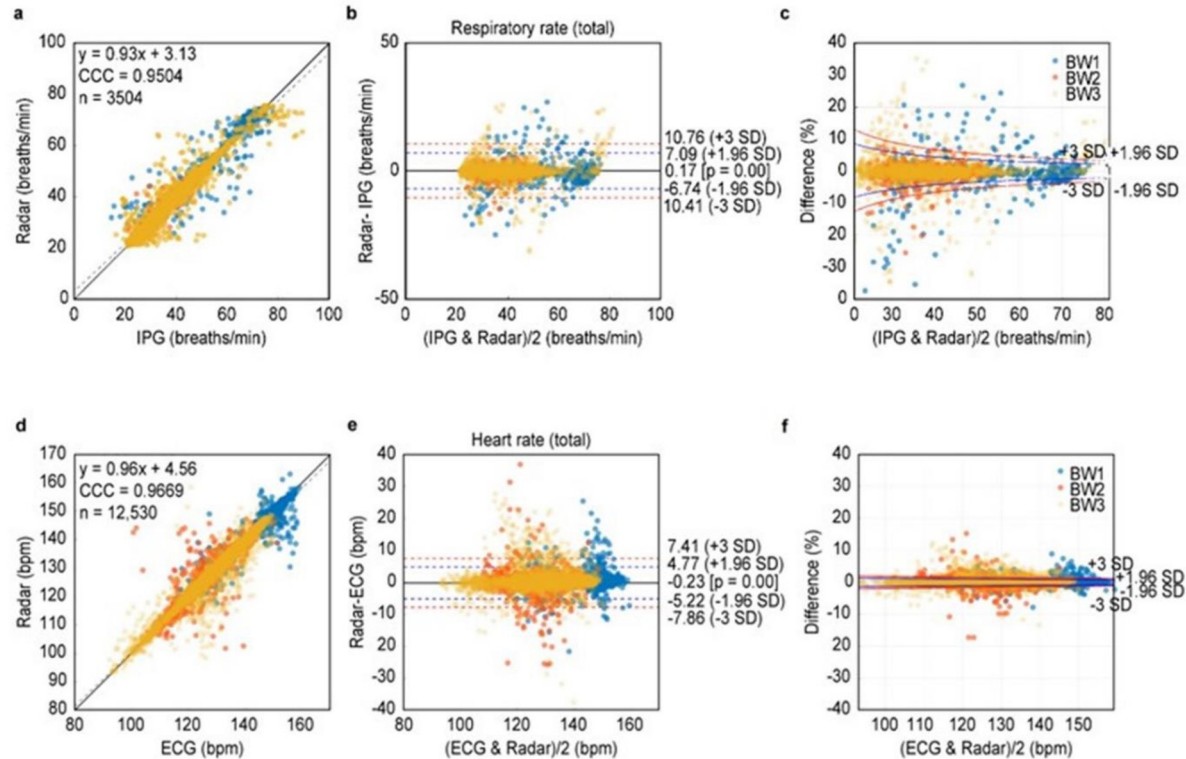

**Fig 5. Agreement for the RR and HR between IR-UWB radar and conventional ECG/IPG.** The $RR_{Rd}$ was highly correlated with the $RR_{IPG}$ (A). BA plot showing that the mean bias between the $RR_{Rd}$ and $RR_{IPG}$ was only 0.17 breaths/minute, which is negligible in clinical practice (B). The width of the 95% LOA of the percent difference between the $RR_{Rd}$ and $RR_{IPG}$ was less than 16.8% of the average RRs at the maximum width and gradually decreased with increasing average RRs (C). The $HR_{Rd}$ was also highly correlated with the $HR_{ECG}$ (D). BA plot showing that the mean bias between the $HR_{Rd}$ and $HR_{ECG}$ was only -0.23 beats/minute, and the width of the 95% LOA was approximately 7.5% of the median average HR (E). The width of the 95% LOA of the percent difference between the $HR_{Rd}$ and $HR_{ECG}$ was less than 2.8% of the average HR at the maximum width and gradually decreased with increasing average HRs (F). BW1, body weight ≤2 kg; BW2, 2< body weight ≤3 kg; BW3, body weight >3 kg.

excellent (CCC 0.97; 95% CI, 0.966–0.968). The Bland-Altman plot shows a very small mean bias (0.23 bpm; 95% CI, -0.18–-0.27; $p<0.001$), though significantly different from zero. The width of the 95% LOA was less than 10% (10.0 bpm) of the average HR at the maximum width. The agreement levels were similar among the BW groups (CCC for BW1 group 0.96 [95% CI 0.96–0.97] vs. CCC for BW2 group 0.94 [95% CI 0.94–0.94] vs. CCC for BW3 0.96 [95% CI 0.96–0.96]). The mean bias of HR measurements was smallest in the BW2 group, while there were no differences in the mean biases of RRs among the 3 body weight groups. The absolute values of the mean biases were <1 in both RR and HR in all 3 groups (Fig 6A, S4 Fig, and S1 Table). The biases were smallest in the HR2 category and RR2 category, whereas the mean biases were significant in all categories in both HR and RR. The radar measured HR and RR more frequently in the low HR and RR and less frequently in the high HR and RR, compared with the conventional monitors (Fig 6B, S4 Fig, and S1 Table). The proportion of the time when the discrepancy during the minimal movements was ≤5 bpm was 95.4% in all neonates and was similar among the 3 BW groups (S1 Table).

## Discussion

For the first time, our innovative wireless technology, IR-UWB radar, successfully detected HR and RR with good signal quality and provided a high degree of accuracy comparable to

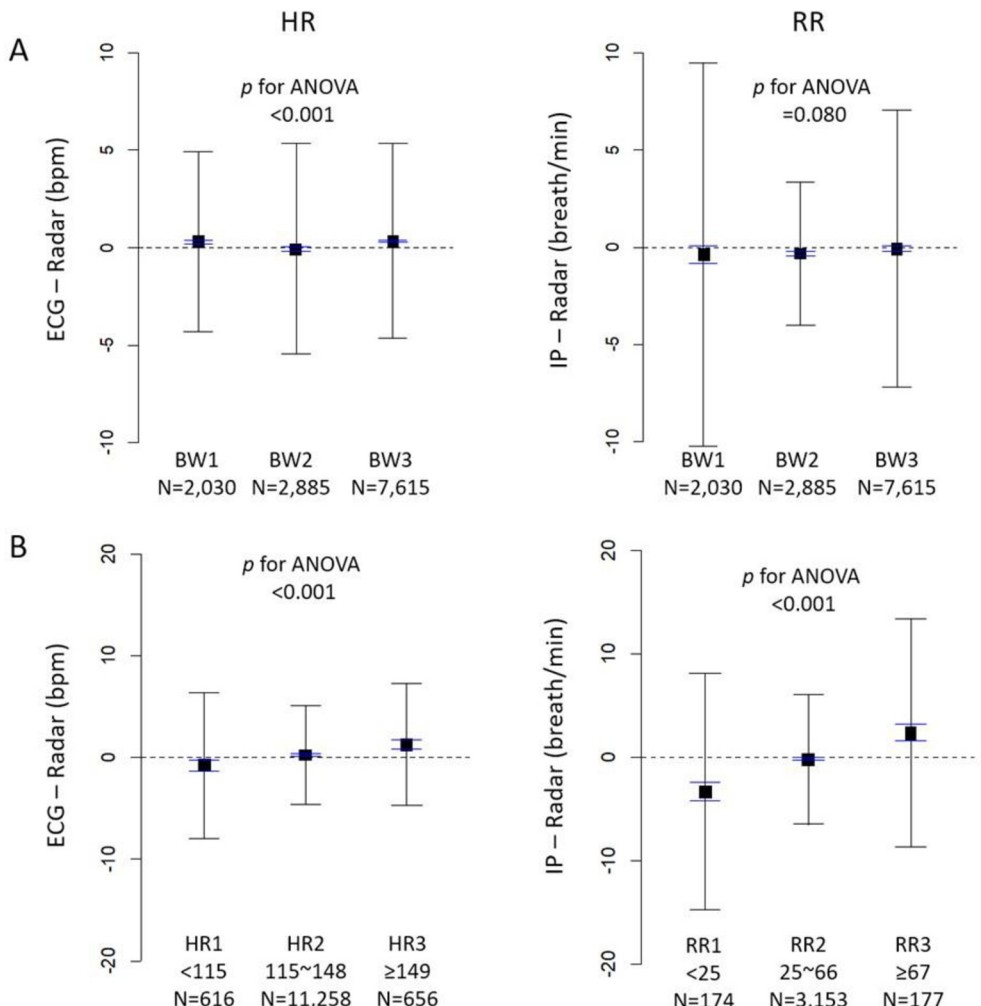

**Fig 6. Agreement of RR and HR between IR-UWB radar and ECG/IPG according to BW, HR and RR.** The means biases (boxes), 95% LOAs (black whiskers) and 95% CI of the mean biases (blue whiskers) were plotted according to BW, HR and RR. A) Although the biases between the two methods are significant for RR in the BW2 group (p<0.001) and for HR in the BW1 (p<0.001) and BW3 (p<0.001) groups in one-sample t-tests, the absolute biases are less than 1 in both HR and RR in all groups. Biases in HR are smallest in the BW2 group while biases in RR are not different among the BW groups. B) Biases are smallest in the HR2 category and RR2 category. The mean biases are significant in all categories. The radar measurements of HR and RR was higher in the low HR and RR range and lower in the high HR and RR range than the conventional measurements.

that of the current standard monitoring technique used in neonates. Although extracting rapid heartbeats with a tiny magnitude of the chest wall of neonates is difficult, we successfully isolated HR from the respiration harmonics using a new algorithm for neonates [19, 20].

The patches and wires of conventional vital monitoring equipment can not only cause misreading of X-ray films but are also a major obstacle to parent-child bonding [2, 9]. Anand and Scalzo [28] suggested that pain, stress and maternal separation of NICU patients have a negative impact on cognitive development [29]. In addition, a study by Chen et al. [30] indicated that stressful conditions, such as repetitive application and removal of patches, adversely affect an infant's well-being and developmental outcomes [31], and the resulting scars may be disfiguring or disabling in 10% of preterm infants [13]. Considering the above findings, a contactless vital sign monitoring technique would be highly desirable.

Accurately monitoring HR in the NICU is very important to clinicians because recurrent episodes of bradycardia may be warning signs of various serious conditions, such as infection or sepsis, respiratory distress, critical arrhythmia, and heart failure in neonates. In particular, as the survival rates of preterm infants increase, alternative non-contact monitoring methods are becoming important. There are only a few explorative pilot studies for monitoring vital signs in the NICU using camera photoplethysmography, laser Doppler vibrometry, piezoelectric sensors, digital stethoscopes, and transcutaneous electromyography, among other technologies [1, 2, 6, 7, 32], but most of the published literature has described small-scale, pilot studies in the developmental phase.

The feasibility and accuracy of the IR-UWB radar have thus far not been studied in the NICU. This study showed that the radar-derived $RR_{Rd}$ and $HR_{Rd}$ correlated perfectly with the $RR_{IPG}$ and $HR_{ECG}$ signals. Our non-contact radar has many advantages in biomedical applications because it is electrodeless, safe, inexpensive, convenient to use, portable, cross-linkable with IoT, and highly compatible with other tubes or catheters attached to neonates [15, 20, 22–24]. Most non-contact sensors are generally susceptible to a baby's motions. In contrast, our IR-UWB radar sensor could integrate movement detection with vital monitoring to reduce false alarms and avoid invalid measurements by automatically cancelling motion-contaminated data. In addition, unlike signals from other non-contact sensors, signals from the radar are not affected by any status of skin, phototherapy, and ambient illumination conditions in the NICU.

Recently, we successfully extracted respiratory signals in neonates [18], and the accuracy of our technology was proven by comparing it with the reference gold standard using a signal filtering algorithm as demonstrated in our preliminary study [20, 33]. However, the accurate calculation of HR for neonates was very challenging and more difficult than respiration detection with our conventional algorithm, mainly because focusing on the neonate's small heart is difficult for the IR-UWB radar, and the HR of the neonate is more than double that of the adult. In addition, the radar suffered from noisy signals relating to the environment and nursing care by the medical staff. Finally, we overcame these obstacles through a modified algorithm suitable for a small human body, and accurate heartbeat information was extracted through a redesigned radar algorithm for neonates. We compared IR-UWB to ECG/IPG, as this is currently the most widely used method for cardiorespiratory monitoring in the NICU. The analysis of HR measurement as well as RR revealed a compatible detection rate and good correlation and agreement between the two methods during minimal movements (Figs 2 and 3).

The overall results suggest that the IR-UWB technique is feasible for the general NICU population. The clinical applicability of the radar could be an attractive option for standard neonatal monitoring. Furthermore, this promising study can be the first and essential step to measure vital signs without contact in neonates and may have important clinical implications as a home monitoring solution for high-risk infants and as a screening tool for serious diseases [5, 20, 34–36].

## Limitations

Vital monitoring through the IR-UWB radar has certain limitations. First, oxygen saturation monitoring is not possible in comparison with an established gold standard. Second, all measurements are obtained in a supine position with a fixed angle, and the device is far from the chest. Third, if subjects have severe bradycardia, with the HR falling to a level similar to the RR, accurate calculation is difficult because the radar measurements overlap. Fourth, recordings with notable body movements are still a challenge.

## Conclusions

This study supports the wireless and electrodeless IR-UWB sensor as an applicable method to collect both HR and RR data for the first time in the NICU. Despite the obvious limitations, expectations for future vital sign monitoring using IR-UWB radar are amplified because of the successful non-contact cardiorespiratory monitoring in neonates. Compared to reference monitoring data, which are widely used in clinical practice, the radar data show similar results. Better hardware and improved algorithms to compensate for neonates' motion are required to increase the robustness of the IR-UWB radar.

## Supporting information

**S1 Checklist. CONSORT 2010 checklist of information to include when reporting a randomised trial**[*]**.**
(DOC)

**S1 Data. To quantify the amount of movement $E_i$, the results of the squared value of each component of the difference between the $n$th received radar signal and the $n−1$ received radar signal over a given threshold $T[k]$ are added together.** This approach is suitable for quantifying and assessing an infant's movements because the more the infant moves, the greater the continuous change in signals received from the radar. This method is also used to measure sedentary movement without any change in position [25].
(DOCX)

**S1 Fig. The spectrum of the radar signals in the frequency domain and the heartbeat waveforms obtained from the radar in real time.** (A) The waveform with the highest magnitude represents the RR frequency location in the spectrum. To extract the HR components, RR harmonic components were removed using the notch filter. Because the magnitudes of RR harmonic components decrease exponentially, the magnitude of the 3rd harmonic component was negligible compared to that of the HR frequency component. (B) Heartbeat waveforms from IR-UWB radar and ECG. The signal waveforms from heartbeats correspond well with the R wave of ECG waveforms.
(DOCX)

**S2 Fig. Representative examples of the HR and RR with notable movement.** The blue and red lines represent measurements from the radar ($RR_{Rd}$, $HR_{Rd}$) and ECG/IPG ($RR_{IPG}$, $HR_{ECG}$) over 2.5 minutes, respectively. The degree of movement is presented with arbitrary units based on the distance from the IR-UWB radar (the lowest panel). The HR and RR values of the two sensors differ significantly when notable movement occurs.
(DOCX)

**S3 Fig. The correlation and accuracy between the $RR_{Rd}$ derived from the IR-UWB radar and the $RR_{IPG}$ from IPG when neonates were stable in the (A) BW1 group, (B) BW2 group, and (C) BW3 group.** The $RR_{Rd}$ shows strong agreement with the $RR_{IPG}$ regardless of body weight. Pearson's correlation coefficient is shown in the left panel of each graph, and the mean difference and the lower and upper limits of agreement by Bland-Altman plots are indicated by the two blue (±1.96 SD) or red (±3 SD) dotted lines in the right panel of each graph.
(DOCX)

**S4 Fig. The correlation and accuracy between the $HR_{Rd}$ derived from the IR-UWB radar and the HRECG from ECG when neonates were stable in the (A) BW1 group, (B) BW2 group, and (C) BW3 group.** The $HR_{Rd}$ also agreed well with the $HR_{ECG}$ regardless of body weight. Pearson's correlation coefficient is shown in the left panel of each graph, and the mean

difference and the lower and upper limits of agreement by Bland-Altman plota are indicated by the two blue (±1.96 SD) or red (±3 SD) dotted lines in the right panel of each graph. (DOCX)

**S1 Table. Agreement between IR-UWB Radar and ECG/IPG for HR and RR measurements when the neonates were stable: Signal matching, correlations, and differences according to the BW groups.**
(DOCX)

**S1 File. Clinical trial protocol original language.**
(DOCX)

**S2 File. Clinical trial protocol translation.**
(DOCX)

## Author Contributions

**Conceptualization:** Jae Yoon Na, Young-Hyo Lim, Seok Hyun Cho, Hyun-Kyung Park.

**Data curation:** Jae Yoon Na, Seung Hyun Kim.

**Funding acquisition:** Sung Ho Cho.

**Investigation:** Seung Hyun Kim, Sung Ho Cho.

**Methodology:** Won Hyuk Lee, Sung Ho Cho.

**Project administration:** Hyun Ju Lee, Seok Hyun Cho, Hyun-Kyung Park.

**Resources:** Hyun Ju Lee.

**Software:** Won Hyuk Lee.

**Supervision:** Yonggu Lee, Hyun-Kyung Park.

**Validation:** Yonggu Lee, Young-Hyo Lim.

**Visualization:** Jae Yoon Na.

**Writing – original draft:** Won Hyuk Lee.

**Writing – review & editing:** Yonggu Lee, Sung Ho Cho, Hyun-Kyung Park.

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
