## [Decision Letter · Decision Letter 0]

20 Aug 2020

PONE-D-20-21820

Feasibility of non-contact cardiorespiratory monitoring using impulse-radio ultra-wideband radar in the Neonatal Intensive Care Unit

PLOS ONE

Dear Dr. Hyun-Kyung Park,

Thank you for submitting your manuscript to PLOS ONE. After careful consideration, we feel that it has merit but does not fully meet PLOS ONE’s publication criteria as it currently stands. Therefore, we invite you to submit a revised version of the manuscript that addresses the points raised during the review process.

We look forward to receiving your revised manuscript.

Kind regards,

Georg M. Schmölzer

Academic Editor

PLOS ONE

Journal Requirements:

2. Please include captions for your Supporting Information files at the end of your manuscript, and update any in-text citations to match accordingly. Please see our Supporting Information guidelines for more information: http://journals.plos.org/plosone/s/supporting-information

Reviewers' comments:

Reviewer's Responses to Questions

**Comments to the Author**

1. Is the manuscript technically sound, and do the data support the conclusions?

Reviewer #1: Yes

2. Has the statistical analysis been performed appropriately and rigorously? 

Reviewer #1: Yes

3. Have the authors made all data underlying the findings in their manuscript fully available?

Reviewer #1: No

4. Is the manuscript presented in an intelligible fashion and written in standard English?

Reviewer #1: Yes

5. Review Comments to the Author

Reviewer #1: This study investigates the feasibility of non-contact cardiorespiratory monitoring using impulse-radio ultra-wideband radar in the Neonatal Intensive Care Unit by comparing the measures with conventional electrocardiography (ECG)/impedance pneumography (IPG). I have several comments and questions for Statistical analysis.

Line 211, the abbreviation of SD must be defined the first time it is used.

Please make it clear what is “the bias between the radar and the conventional measurements”. Does such “bias” equal to the difference between the radar and the conventional measurements?

I do not see any reports in this manuscript about intra-individual variances and inter-individual variances. How would you incorporate repeated measures in one-way ANOVA?

According to data shown in S1 table, in Figure 5, the titles of the two boxes should be exchanged.

In figure 5, what do those vertical bars represent? One sample t-tests is not necessary because small difference from zero will get significant result when the sample size is large. To reflect the mean bias, 95% CI may be used.

6. PLOS authors have the option to publish the peer review history of their article (what does this mean?). If published, this will include your full peer review and any attached files.

Reviewer #1: No

---

## [Author Response · Author response to Decision Letter 0]

30 Aug 2020

Reviewer #1: This study investigates the feasibility of non-contact cardiorespiratory monitoring using impulse-radio ultra-wideband radar in the Neonatal Intensive Care Unit by comparing the measures with conventional electrocardiography (ECG)/impedance pneumography (IPG). I have several comments and questions for Statistical analysis.

1. Line 211, the abbreviation of SD must be defined the first time it is used.

: Thanks for your comment. We have defined the term of SD in Line 204 at first instead. 

2. Please make it clear what is “the bias between the radar and the conventional measurements”. Does such “bias” equal to the difference between the radar and the conventional measurements?

: Thanks for your detailed comment. We want to talk about the difference between the radar and the conventional measurement, so we have changed “bias” to “differences” as your comment in Line 211.

3. I do not see any reports in this manuscript about intra-individual variances and inter-individual variances. How would you incorporate repeated measures in one-way ANOVA?

: Thank you for your deep interest in statistically analysis. We used CCC to check the agreement of the measurements of the IR-UWB radar sensor and conventional measurements. Various statistical methods were considered to analyze the agreement between both methods, but since one-way ANOVA was not used, the content has been removed from the manuscript.

4. According to data shown in S1 table, in Figure 5, the titles of the two boxes should be exchanged.

: Thank you for your comments. As your comment, it is correct that the two boxes in Figure 5 have been exchanged. Including the comment below, we would revise Figure 5 accordingly.

5. In figure 5, what do those vertical bars represent? One sample t-tests is not necessary because small difference from zero will get significant result when the sample size is large. To reflect the mean bias, 95% CI may be used.

: Thank you for your comments. Figure 5 shows the t-test results for the difference between heart rate and respiratory rate of the IR-UWB radar sensor and the conventional measurements. Vertical bar means CI, and the figure is incorrect, so we corrected the figure rightly.

During the overall review of the manuscript submitted, there was a mistake in the radar device name used for data collection. We changed the product name to XK200 (Xandar Kardian, Delaware, USA) in Line 126 and Figure 1. Also, the website address related to FCC certification has expired, so we have attached a new address and revised it in Line 133. We’re sorry for such a mistake.

---

## [Decision Letter · Decision Letter 1]

30 Oct 2020

PONE-D-20-21820R1

Feasibility of non-contact cardiorespiratory monitoring using impulse-radio ultra-wideband radar in the Neonatal Intensive Care Unit

PLOS ONE

Dear Dr. Hyun-Kyung Park,

Thank you for submitting your manuscript to PLOS ONE. After careful consideration, we feel that it has merit but does not fully meet PLOS ONE’s publication criteria as it currently stands. Therefore, we invite you to submit a revised version of the manuscript that addresses the points raised during the review process.

We look forward to receiving your revised manuscript.

Kind regards,

Georg M. Schmölzer

Academic Editor

PLOS ONE

Reviewers' comments:

Reviewer's Responses to Questions

**Comments to the Author**

1. If the authors have adequately addressed your comments raised in a previous round of review and you feel that this manuscript is now acceptable for publication, you may indicate that here to bypass the “Comments to the Author” section, enter your conflict of interest statement in the “Confidential to Editor” section, and submit your "Accept" recommendation.

Reviewer #2: (No Response)

2. Is the manuscript technically sound, and do the data support the conclusions?

Reviewer #2: Yes

3. Has the statistical analysis been performed appropriately and rigorously? 

Reviewer #2: Yes

4. Have the authors made all data underlying the findings in their manuscript fully available?

Reviewer #2: Yes

5. Is the manuscript presented in an intelligible fashion and written in standard English?

Reviewer #2: Yes

6. Review Comments to the Author

Reviewer #2: Review of: Feasibility of non-contact cardiorespiratory monitoring using impulse-radio ultrawideband radar in the Neonatal Intensive Care Unit

Lee WH et al. PONE-D-20-21820R1

Lee et al report the ‘next step’ in development of a non-contact cardiorespiratory monitoring technology for heart rate and respiratory rate monitoring of newborn infants. This follows their pre-clinical report published last year (Park JY et al, https://www.ncbi.nlm.nih.gov/pmc/articles/PMC6695386/ ). It is well written, and provides information on a promising technology. They do a good job in describing the limitations and next steps before this technology moves closer to clinical application. As a non-statistician, my most significant question for this or a follow-on question is how reliable, valid and accurate the new technology is relative to the gold standard when analyzing high and low Heart Rates and respiratory rates? Can alarms be triggered? Would they be triggered too often, not often enough, or at an expected rate? Other minor comments/questions follow.

Introduction.

The authors might want to identify citations to justify the statement: “Repetitive replacement of electrodes and the twining wires around the arm or leg cause skin damage, infections due to skin layer breakdown, permanent scars, and circulatory disturbances, particularly in premature infants with fragile skin. There may even be a risk of hypothermia during procedures, which could cause circulatory disturbances, particularly in premature infants.

In addition to reference 2, the authors may want to cite a more recent reference on the impact of all the current leads in use on family interactions with their neonates (Bonner O, et al. 'There were more wires than him': the potential for wireless patient monitoring in neonatal intensive care. BMJ Innov. 2017;3(1):12-18. doi:10.1136/bmjinnov-2016-000145).

METHODS

Very minor, but in the location for MATLAB, the authors list the company location as “MathWorks, New York, MA, USA”). I think the headquarters of the company is Natick, MA, USA.

Very minor, line 176 of the revised manuscript w/ tracked changes, one word is misspelled: “...want to increase the Frames per sencod (FPS) of the radar to increase the quality of...” should be “...want to increase the Frames per second (FPS) of the radar to increase the quality of...”.

I am not that statistically savvy, but would be interested in how well the correlations hold up at the lower and upper extremes of HR and RR. Are there specific statistical tests for measurement comparisons w/ gold standard technologies that accentuate the evaluation of the extremes? Maybe something like Figure 4 with HR and RR low, high and middle rates in place of the BW groupings would give a visual representation that I’m sure a wise statistician could translate more quantitatively.

DISCUSSION

It’s one really good thing to pick up the normal range for HR and RR. How does the radar technology perform in picking up apnea and bradycardia? Could the authors in this (or a subsequent) paper tell the reader specifically about correlations between two techniques in alarms and abnormal readings on the low and high ends of both HR and RR?

7. PLOS authors have the option to publish the peer review history of their article (what does this mean?). If published, this will include your full peer review and any attached files.

Reviewer #2: No

---

## [Author Response · Author response to Decision Letter 1]

25 Nov 2020

Review Comments to the Author

Reviewer #2: Review of: Feasibility of non-contact cardiorespiratory monitoring using impulse-radio ultrawideband radar in the Neonatal Intensive Care Unit

Lee WH et al. PONE-D-20-21820R1

Lee et al report the ‘next step’ in development of a non-contact cardiorespiratory monitoring technology for heart rate and respiratory rate monitoring of newborn infants. This follows their pre-clinical report published last year (Park JY et al, https://www.ncbi.nlm.nih.gov/pmc/articles/PMC6695386/ ). It is well written, and provides information on a promising technology. They do a good job in describing the limitations and next steps before this technology moves closer to clinical application. As a non-statistician, my most significant question for this or a follow-on question is how reliable, valid and accurate the new technology is relative to the gold standard when analyzing high and low Heart Rates and respiratory rates? Can alarms be triggered? Would they be triggered too often, not often enough, or at an expected rate? Other minor comments/questions follow.

Introduction.

The authors might want to identify citations to justify the statement: “Repetitive replacement of electrodes and the twining wires around the arm or leg cause skin damage, infections due to skin layer breakdown, permanent scars, and circulatory disturbances, particularly in premature infants with fragile skin. There may even be a risk of hypothermia during procedures, which could cause circulatory disturbances, particularly in premature infants.

In addition to reference 2, the authors may want to cite a more recent reference on the impact of all the current leads in use on family interactions with their neonates (Bonner O, et al. 'There were more wires than him': the potential for wireless patient monitoring in neonatal intensive care. BMJ Innov. 2017;3(1):12-18. doi:10.1136/bmjinnov-2016-000145).

: Thanks for your comments. As you advised, we have added the above reference to our manuscript (Reference 9). 

METHODS

Very minor, but in the location for MATLAB, the authors list the company location as “MathWorks, New York, MA, USA”). I think the headquarters of the company is Natick, MA, USA.

: Thank you for your comments. We have modified the location of the company’s headquarter in Line 131.

Very minor, line 176 of the revised manuscript w/ tracked changes, one word is misspelled: “...want to increase the Frames per sencod (FPS) of the radar to increase the quality of...” should be “...want to increase the Frames per second (FPS) of the radar to increase the quality of...”.

: Thank you for your comments. The corresponding misspelled words was modified and corrected in line 178.

I am not that statistically savvy, but would be interested in how well the correlations hold up at the lower and upper extremes of HR and RR. Are there specific statistical tests for measurement comparisons w/ gold standard technologies that accentuate the evaluation of the extremes? Maybe something like Figure 4 with HR and RR low, high and middle rates in place of the BW groupings would give a visual representation that I’m sure a wise statistician could translate more quantitatively.

: Thank you for the comment. We have compared the bias levels among the 3 body weight groups through a graphical presentation and one sample t-test in the last manuscript. However, the more proper method to compare the bias levels among 3 groups would be one-way ANOVA (analysis of variance) with/without post-hoc tests such as Bonferroni or Tukey method. Therefore, we included the results of ANOVA tests among the 3 body weight groups in the revised manuscript (Figure 5A). 

In response to your comment, we think that the bias levels between the radar and the conventional methods could also be compared among categories divided using HR and RR. Lin’s concordance correlation coefficients should not be used for the comparison because there were no established methods for the correction of the range restriction problem that must follow the categorization of the variable through dividing the data range. However, comparisons of the biases between two measurements could bypass this problem. We defined HR and RR > upper 5% and < lower 5% as the extreme values and divided the measurements into 3 categories as follow; Category 1: <5%, Category 2: 5~95% and Category 3: >95%. HRs were divided at 115 and 149 bpm and RRs were divided at 25 and 67 breaths/minute. We compared the bias levels between the two measurement methods among these 3 groups for both HR and RR using graphical presentations and one-any ANOVA (Figure 5B).

The biases between the two measurement methods were smaller in the category 2 (5%~95%) in both HR and RR and the radar measured HR and RR more frequently in the low HR and RR range and less frequently in the high HR and RR range.

In accordance with your comment, we revised the manuscript including these analysis results as Figure 5B. 

DISCUSSION

It’s one really good thing to pick up the normal range for HR and RR. How does the radar technology perform in picking up apnea and bradycardia? Could the authors in this (or a subsequent) paper tell the reader specifically about correlations between two techniques in alarms and abnormal readings on the low and high ends of both HR and RR?

: Thank you for your comments. We are developing algorithms to detect apnea, arrhythmia and bradycardia using radar technology. However, in the NICU environment, the treatment comes first when the symptoms appear, so the low number of times apnea, arrhythmia, and bradycardia have occurred is early to verify the accuracy of detecting them. We will conduct an experiment to accurately detect and verify the accuracy of this part in further research.

Plsease see attched "(2nd) Response to Reviewers" file.

---

## [Editor Report · Decision Letter 2]

1 Dec 2020

Feasibility of non-contact cardiorespiratory monitoring using impulse-radio ultra-wideband radar in the Neonatal Intensive Care Unit

PONE-D-20-21820R2

Dear Dr. Hyun-Kyung Park,

We’re pleased to inform you that your manuscript has been judged scientifically suitable for publication and will be formally accepted for publication once it meets all outstanding technical requirements.

Kind regards,

Georg M. Schmölzer

Academic Editor

PLOS ONE
---

## [Editor Report · Acceptance letter]

15 Dec 2020

PONE-D-20-21820R2 

Feasibility of non-contact cardiorespiratory monitoring using impulse-radio ultra-wideband radar in the Neonatal Intensive Care Unit 

Dear Dr. Park:

I'm pleased to inform you that your manuscript has been deemed suitable for publication in PLOS ONE. Congratulations! Your manuscript is now with our production department. 

Kind regards, 

on behalf of

Dr. Georg M. Schmölzer 

Academic Editor

PLOS ONE